# Identification of Transcriptional Markers and microRNA–mRNA Regulatory Networks in Colon Cancer by Integrative Analysis of mRNA and microRNA Expression Profiles in Colon Tumor Stroma

**DOI:** 10.3390/cells8091054

**Published:** 2019-09-08

**Authors:** Md. Nazim Uddin, Mengyuan Li, Xiaosheng Wang

**Affiliations:** 1Biomedical Informatics Research Lab, School of Basic Medicine and Clinical Pharmacy, China Pharmaceutical University, Nanjing 211198, China (M.N.U.) (M.L.); 2Cancer Genomics Research Center, School of Basic Medicine and Clinical Pharmacy, China Pharmaceutical University, Nanjing 211198, China; 3Big Data Research Institute, China Pharmaceutical University, Nanjing 211198, China

**Keywords:** colon tumor stroma, transcriptional markers, prognostic factors, miRNA–mRNA regulatory networks, tumor immune microenvironment

## Abstract

The aberrant expression of microRNAs (miRNAs) and genes in tumor microenvironment (TME) has been associated with the pathogenesis of colon cancer. An integrative exploration of transcriptional markers (gene signatures) and miRNA–mRNA regulatory networks in colon tumor stroma (CTS) remains lacking. Using two datasets of mRNA and miRNA expression profiling in CTS, we identified differentially expressed miRNAs (DEmiRs) and differentially expressed genes (DEGs) between CTS and normal stroma. Furthermore, we identified the transcriptional markers which were both gene targets of DEmiRs and hub genes in the protein–protein interaction (PPI) network of DEGs. Moreover, we investigated the associations between the transcriptional markers and tumor immunity in colon cancer. We identified 17 upregulated and seven downregulated DEmiRs in CTS relative to normal stroma based on a miRNA expression profiling dataset. Pathway analysis revealed that the downregulated DEmiRs were significantly involved in 25 KEGG pathways (such as TGF-β, Wnt, cell adhesion molecules, and cytokine–cytokine receptor interaction), and the upregulated DEmiRs were involved in 10 pathways (such as extracellular matrix (ECM)-receptor interaction and proteoglycans in cancer). Moreover, we identified 460 DEGs in CTS versus normal stroma by a meta-analysis of two gene expression profiling datasets. Among them, eight upregulated DEGs were both hub genes in the PPI network of DEGs and target genes of the downregulated DEmiRs. We found that three of the eight DEGs were negative prognostic factors consistently in two colon cancer cohorts, including *COL5A2*, *EDNRA*, and *OLR1*. The identification of transcriptional markers and miRNA–mRNA regulatory networks in CTS may provide insights into the mechanism of tumor immune microenvironment regulation in colon cancer.

## 1. Background

The tumor microenvironment (TME) is complex “ecosystems” which comprises many different cell types including stromal cells [1]. The functions of stromal cells are often altered in the TME, and the tumor stroma is associated with tumor cellular migration, neoangiogenesis, immunosurveillance evasion, and drug resistance [2]. Colorectal cancer (CRC) is one of the leading causes of cancer mortality worldwide [3]. Transcriptional signatures of CRC stromal cells have been associated with poor prognosis in CRC [4,5,6]. MicroRNAs (miRNAs) play a critical role in modulating the TME in CRC [7]. Bullock et al. showed that stromal and epithelial miRNAs play key roles during CRC progression [8]. Hiyoshi et al. revealed that the elevated expression of miRNAs-34b and -34c in stromal tissues is associated with poor prognosis in colon cancer [9]. These prior studies demonstrated the significant contribution of aberrantly expressed miRNAs in colorectal tumor stroma to CRC growth, invasion, and metastasis.

The miRNA–mRNA interaction networks in cancers have been widely explored. For example, Taguchi et al. identified the potential miRNA–mRNA interactions within multiple cancer types, including colorectal/colon cancer, hepatocellular carcinoma, non-small cell lung cancer, esophageal squamous cell cancer, prostate cancer, and breast cancer [10]. Yang et al. constructed miRNA and mRNA integrative networks as key regulators in colon adenocarcinoma (COAD) [11]. Li et al. built microRNA–mRNA interactions in human colon cancer [12]. Paul et al. identified potential miRNA–mRNA regulatory modules in CRC based on miRNA and mRNA expression data [13]. Sells et al. showed that the tissue kallikrein-related peptidase 6 (KLK6) mediated specific microRNA–mRNA regulatory networks associated with colon cancer invasion [14]. Izadi et al. investigated the conserved mRNA–miRNA interactions in colon and lung cancers [15]. Besides CRC, other cancer-associated miRNA–mRNA interactions have been explored. For example, Pham et al. identified miRNA–mRNA regulatory relationships which were consistent across different breast cancer subtypes [16]. Lou et al. constructed an miRNA–mRNA regulatory network potentially associated with glioblastoma multiforme [17]. These prior studies demonstrated that the miRNA–mRNA regulatory networks play crucial roles in the pathogenesis of cancer. However, the investigation of miRNA–mRNA regulatory networks in the TME, such as tumor stroma, remains lacking.

In this study, using two gene and miRNA expression profiling datasets, we identified differentially expressed miRNAs (DEmiRs) and differentially expressed genes (DEGs) between colon tumor stroma (CTS) and normal stroma. For these DEmiRs, we identified their associated pathways, target genes, and upstream transcriptional regulators. We performed these bioinformatics analyses using several publicly available tools or databases, including miRNet [18], Network Analyst [19], FunRich [20], DIANA-mirPath [21], TargetScan [22], GEPIA [23], PrognoScan [24], TIMER [25], STRING [26], and DGIdb [27]. Furthermore, we evaluated the correlations between the expression levels of the significant genes targeted by DEmiRs and antitumor immune signatures in CTS. Finally, we established a miRNA–mRNA regulatory network potentially associated with colon cancer progression, prognosis, and immune evasion.

## 2. Methods

### 2.1. Datasets

We obtained two CTS gene (or miRNA) expression profiling datasets: GSE31279 (sample size n = 20) [28] and GSE35602 (*n* = 17) [29] by searching the NCBI gene expression omnibus (GEO) database (https://www.ncbi.nlm.nih.gov/geo/) using the keywords “colon cancer”, “stroma”, “colon tumor stroma”, and “tumor stroma”. GSE31279 and GSE35602-GPL6480 (*n* = 17) were used for identifying DEGs and GSE35602-GPL8227 (*n* = 17) [29] for identifying DEmiRs between CTS and normal stroma. In addition, we used three CRC gene expression profiling datasets (GSE17536 [30,31], GSE14333 [32], and TCGA COAD) [33] for survival analyses. The TCGA COAD dataset was downloaded from the TCGA data portal (https://portal.gdc.cancer.gov/). The stromal fibroblast cell gene expression profiling dataset GSE46824 (*n* = 34) [34] was downloaded from GEO.

### 2.2. Identification of DEGs and DEmiRs between CTS and Normal Stroma

We used Network Analyst [19] to identify the DEGs between CTS and normal stroma by a meta-analysis of two CTS gene expression profiling datasets (GSE31279 and GSE35602). Each dataset was normalized by quantile normalization. The batch effects from both datasets were removed using the ComBat method [35]. We used the R package “limma” to identify the DEGs between CTS and normal stroma, and the Cochran′s combination test to perform the meta-analysis [36]. The false discovery rate (FDR) [37] was used to adjust for multiple tests. We identified the DEGs using a threshold of absolute combined effect size (ES) > 1.24 and FDR < 0.05. We used miRNet [18] to identify DEmiRs between CTS and normal stroma by analyzing the preprocessed GSE35602-GPL8227. The dataset was normalized by quantile normalization, and the R package “limma” was utilized to identify the DEmiRs between CTS and normal stroma using a threshold of |log2(FC)| > 1.0 and adjusted *p*-value < 0.05. The “FC” denotes the fold change of mean expression levels between two compared groups.

### 2.3. Identification of Upstream Transcription Factors and Target Genes of DEmiRs 

The upstream transcription factors (TFs) regulating DEmiRs were predicted using FunRich [28], a tool for analyzing functional enrichment and interaction network of genes and proteins. The most significant (with the smallest *p*-values) 10 upstream TFs for the upregulated and downregulated DEmiRs were identified, respectively. We used miRNet [18], DIANA-microT-CDS [38], and DIANA-TargetScan [22] to identify target genes of DEmiRs. The online tool “Calculate and draw custom Venn diagrams” (http://bioinformatics.psb.ugent.be/webtools/Venn/) was used to identify common genes between the target genes of DEmiRs and the DEGs between CTS and normal stroma. 

### 2.4. KEGG Pathway Enrichment Analysis of DEmiRs

We identified KEGG [39] pathways that were significantly associated with the DEmiRs by DIANA-mirPath (version 3) [21]. In this tool, we chose to analyze the validated miRNA–gene interactions using archive in microT web server v5.0 [38]. A threshold of adjusted *p*-value < 0.05 (Fisher’s exact test) was used to define the statistical significance.

### 2.5. Construction of Protein–Protein Interaction (PPI) Network of DEGs 

We constructed a PPI network of the DEGs between CTS and normal stroma by STRING [26]. The hub genes (with no less than 10 edges connected to other nodes) in the PPI network were identified using the node explorer module of Network Analyst [19].

### 2.6. Comparisons of the Expression Levels of Hub Genes between Colon Cancer and Normal Tissue and Between Colon Tumor Stromal Fibroblast and Normal Fibroblast

We compared the expression levels of the hub genes in the PPI network which were also the targets of the DEmiRs between TCGA COAD and normal tissue, and between colon tumor stromal fibroblast and normal fibroblast. A threshold of Student′s *t* test *p*-value < 0.05 and |FC| > 1 was utilized to denote the statistical significance.

### 2.7. Associations of the Expression Levels of the Hub Genes Targeted by DEmiRswith Survival Prognosis in Colon Cancer

We analyzed the associations of the expression levels of the hub genes targeted by DEmiRs with survival prognosis in colon cancer using PrognoScan [24] and GEPIA [23]. The log-rank test *p*-value < 0.05 denoted the statistical significance.

### 2.8. Associations of the Expression Levels of the Hub Genes Targeted by DEmiRs with Immune Signature Enrichment Levels in Colon Cancer

We analyzed the correlations of the expression levels of three hub genes (*COL5A2*, *EDNRA*, and *OLR1*) with the abundance of immune signatures in TCGA COAD using TIMER [25]. Moreover, we compared the ratios of pro-inflammatory cytokines to anti-inflammatory cytokines between the colon cancers with high expression levels of the hub genes (expression levels > median) and the colon cancers with low expression levels of the hub genes (expression levels < median) in the TCGA COAD dataset using Student′s *t* test. We defined the ratio of pro-/anti-inflammatory cytokines in a tumor sample as the ratio of average expression levels (base-2 log transformed) of their marker genes. The pro-inflammatory cytokines represent the immune-stimulatory signature with marker genes *IFNG*, *IL-1A*, *IL-1B*, and *IL-2*, and the anti-inflammatory cytokines represent the immune-inhibitory signature with marker genes *IL-4*, *IL-10*, *IL-11*, and *TGFB1* [40].

### 2.9. Identification of Food and Drug Administration (FDA)-Approved Drug-Hub Gene Interaction

We identified the drugs that target the hub genes using DGIdb [27]. DGIdb collects drug-gene interaction data from 30 disparate sources, including ChEMBL, DrugBank, Ensembl, NCBI Entrez, PharmGKB, PubChem, clinical trial databases, and literature in NCBI PubMed. The drug-gene interactions supported by at least one database and/or PubMed reference were identified. From the identified drug-gene interactions, we selected the drugs that have been approved by the FDA.

## 3. Results

### 3.1. Identification of DEmiRs and Their Target DEGs

We identified seven downregulated and 17 upregulated DEmiRs in CTS relative to colon normal stroma (Table 1). A literature review showed that all seven miRNAs downregulated in CTS have been associated with colon cancer and other cancer types. For example, hsa-mir-135b-5pwas significantly downregulated in CTS with the highest fold change of expression levels (FDR = 0.009, FC > 7). This miRNA has been associated with gastric carcinogenesis [41]. Hsa-miR-214-3p was significantly downregulated in CTS (FDR = 0.02, FC > 3) and was related to colon cancer risk [42]. Hsa-mir-224-5p was related to the pathogenesis of COAD [11]. Hsa-mir-495-3p might play an important role in the tumorigenesis of glioma [43]. Interestingly, our results showed that hsa-miR-21-5p, hsa-mir-21-3p, and hsa-mir-409-3p were downregulated in CTS, while their expression levels were increased in CRC tissues [44,45,46].

Many of the significantly upregulated DEmiRs in CTS have been associated with CRC. For example, the expression levels of hsa-mir-375 were 5.74 times higher in CTS than in normal stroma (*p* = 0.03). This miRNA was also differentially expressed between CRC and normal tissue [47]. Hsa-mir-192-3p was enriched in different extracellular vesicle subtypes derived from CRC cell lines [48]. The elevated expression of hsa-mir-215-5pwas associated with CRC metastasis [49]. Hsa-mir-378a-3pwas differentially expressed between human colorectal carcinoma and normal colonic mucosa [50]. The analysis of circulating miRNA expression profiles revealed that hsa-mir-498was upregulated in colon cancer patients [46].

Therefore, a majority of the aberrantly expressed DEmiRs in CTS relative to normal stroma identified by the bioinformatics approach have been related to the pathogenesis of colon cancer.

### 3.2. Identification of Upstream TFs and Genes Significantly Associated with the DEmiRs

We used FunRich [20] to identify upstream TFs regulating the DEmiRs. For the upregulated DEmiRs, the most significant 10 upstream TFs included EGR1, SP1, SP4, NKX6-1, POU2F1, MEF2A, RREB1, ZFP161, NFIC, and ONECUT1 (Table 2). For the downregulated DEmiRs, the most significant 10 TFs included SP4, SP1, EGR1, HOXA5, PDX1, KLF7, RORA, POU2F1, TCF3, and FOXD3 (Table 2). Previous studies have revealed that most of the upstream TFs are related to CRC. For example, SP1 regulating most of the upregulated DEmiRs has been associated with colon tumor adhesion, migration and invasion [51]. SP1 and SP4 are essential for colon cancer cells and knockdown of them may induce apoptosis in cancer cells [52]. EGR1can promote proliferation of colon cancer cells via the EGR1/AE2/P16/P-ERK signaling pathway [53]. POU2F1 is involved in the regulation of colon malignancy [54]. TCF3 has been identified as an upstream TF of CTS-specific transcriptional signatures [55]. FOXD3 plays an important role in inhibiting colon cancerogenesis by regulating EGFR/Ras/Raf/MEK/ERK signaling pathway [56].

Furthermore, we predicted target genes of DEmiRs using three different tools (miRNet [18], MicroT-CDS [38], and TargetScan [22]) (Appendix A), and obtained the common target genes predicted by the three tools. We identified 2056 DEmiR-gene regulation relationships based on these DEmiRs using miRNet [18], MicroT-CDS [38] and TargetScan [22]. In total, 1021 and 620 genes were predicted as targets of the upregulated and downregulated DEmiRs, respectively.

### 3.3. Identification of Pathways Significantly Associated with DEmiRs

We identified 25 KEGG [39] pathways significantly associated with the downregulated DEmiRs in CTS using DIANA-miRPath [21]. These pathways are mainly involved in cancer (pathways in cancer, pancreatic cancer, and proteoglycans in cancer), cellular development (cell adhesion molecules, Wnt, TGF-β, FoxO, adherens junction, Gap junction, and ErbB pathways), immune regulation (cytokine–cytokine receptor interaction), and metabolism (glycosphingolipid biosynthesis-lacto and neolacto series, biosynthesis of unsaturated fatty acids, and ubiquitin mediated proteolysis) (Appendix A). Previous studies have shown that some of these pathways are significantly associated with colon cancer or CTS. For example, the Wnt signaling, cell adhesion molecules, adherens junction, gap junction, and cytokine–cytokine receptor interaction pathways have been associated with CTS [6]. Genetic variation in the TGF-β signaling pathway may lead to an increased risk of CRC [57]. ErbB signaling pathway is involved in cell proliferation, invasion, metastasis, and neoangiogenesis in colon cancer [58]. Moreover, we identified 10 KEGG pathways that were significantly associated with the upregulated DEmiRs in CTS (Appendix A). Many of these pathways have been associated with colon cancer. For example, the extracellular matrix (ECM)-receptor interaction pathway is highly enriched in CTS [6]. Dysregulation of proteoglycans is associated with colon cancer prognosis [59].

### 3.4. Identification of Differentially Expressed Hub Genes Targeted by DEmiRs

We identified 460 DEGs between CTS and normal stroma by a meta-analysis of two GEO microarray datasets (GSE31279 andGSE35602). These DEGs included 160 downregulated and 300 upregulated genes in CTS (Appendix A). Based on the 460 DEGs, we constructed a PPI network to identify hub genes (Appendix A) and found eight hub genes (degree ≥ 10) targeted by the DEmiRs (Figure 1A). All the eight genes (*COL5A2*, *EDNRA*, *OLR1*, *TGFBI*, *MET*, *TNFRSF11B*, *TWIST1*, and *WNT5A*) were upregulated in CTS and were targeted by the downregulated DEmiRs (Figure 1B). *COL5A2* is predominantly expressed in COAD stroma [60]. A meta-analysis of gene expression profiles in 950 cancer cell lines revealed that *OLR1* was upregulated in 20% of CRC cells [61].

Interestingly, we found that all eight hub genes (*COL5A2*, *EDNRA*, *TGFBI*, *MET*, *OLR1, TNFRSF11B, TWIST1,* and *WNT5A*) were also upregulated in TCGA COAD samples versus normal samples (Student′s *t* test, *p* < 0.001) (Figure 2). It indicates that stromal components may have a contribution to COAD transcriptome. However, none of the hub genes showed significant expression differences between colon tumor stromal fibroblast and normal fibroblast (Appendix A), suggesting that these hub genes are specifically expressed in other CTS cells.

#### 3.4.1. The Hub Genes Are Negative Prognostic Factors in CRC

Survival analyses using Prognoscan [24] showed that three hub genes (*COL5A2*, *EDNRA*, and *OLR1*) had a negative expression correlation with survival prognosis (overall survival (OS) and/or disease-free survival (DFS)) in CRC (Figure 3A). Moreover, the elevated expression of *COL5A2*, *EDNRA*, *TGFBI*, and *OLR1* was associated with a worse DFS prognosis in TCGA COAD (Figure 3B). Previous studies have shown that some of the hub genes are associated with prognosis in colon cancer. For example, the depressed expression of *EDNRA* is associated with a better prognosis in colon cancer [62]. Taken together, these data demonstrate that the upregulated hub genes regulated by the downregulated DEmiRs in CTS are negative prognostic factors in colon cancer.

#### 3.4.2. The Elevated Expression of Hub Genes Is Associated with the Immunosuppressive TME in Colon Cancer

The tumor-infiltrating lymphocytes (TILs) level is an independent predictor of survival in CRC [63]. We analyzed the correlations between the expression levels of three hub genes and the levels of TILs in TCGA COAD. The three hub genes included *COL5A2*, *EDNRA*, and *OLR1* which were identified as negative prognostic factors in both TCGA COAD [33] and two microarray CRC cohorts (GSE17536 [30,31] and GSE14333) [32]. Interestingly, we found that the elevated expression of the three genes was consistently correlated with the upregulation of immunosuppressive signatures in colon cancer, such as tumor-associated macrophages (TAMs), M2 macrophages, regulatory T cells (Tregs), and T cell exhaustion (Table 3). Moreover, the expression levels of the three genes exhibited significant positive correlations with the expression levels of immune-inhibitory marker genes, including *PD-L1*, *PD-L2*, *TGFB1*, and *TGFBR1* (Figure 4A). Interestingly, we found that the ratios of pro-/anti-inflammatory cytokines were significantly lower in the colon cancers highly expressing the hub genes (expression levels > median) than in those lowly expressing the hub genes (expression levels < median) in the TCGA COAD cohort (Student’s *t* test, *p* < 0.001) (Figure 4B). Altogether, these results suggest that the elevated expression of these hub genes is associated with the immunosuppressive TME in colon cancer that could explain their oncogenic function to a certain degree.

### 3.5. Identification of Candidate Drugs Targeting Hub Genes

Using DGIdb [27], we identified eight FDA-approved drugs that potentially target the protein products of two hub genes (*EDNRA* and *COL5A2*) (Table 4). Table 4 shows that the drugs targeting EDNRA are mostly antagonists. EDNRA (Endothelin receptor) antagonists ambrisentan, bosentan, and macitentan have been evaluated as hepatobiliary transporter inhibitors and substrates in sandwich-cultured human hepatocytes [65]. The efficacy of macitentan, a dual endothelin receptor antagonist, has been evaluated in treating the brain metastases of breast and lung cancers in mice [66]. The use of macitentan, a dual endothelin receptor antagonist, in combination with temozolomide, may lead to glioblastoma regression and long-term survival in mice [67]. Nevertheless, to our knowledge, inhibition of EDNRA or COL5A2 has not been tested for colon cancer treatment. Our data suggest that these genes could be promising targets for development of anticancer drugs in treating colon cancer patients.

## 4. Discussion

For the first time, we built the miRNA–mRNA regulatory network based on the aberrantly expressed miRNAs and genes in CTS. We identified seven downregulated and 17 upregulated DEmiRs in CTS relative to normal stroma by a meta-analysis. A recent study has revealed that TFs can modulate the expression of miRNAs, and they have a synthetic effect on the expression of their common target genes in colon cancer [68]. Thus, we identified upstream TFs that potentially regulated the DEmiRs. Notably, SP1, a zinc finger TF, potentially regulated a majority of the DEmiRs. Pathway enrichment analysis revealed that these DEmiRs were significantly associated with the pathways involved in cancer, cellular development, immune regulation, and metabolism. Furthermore, we identified DEGs between CTS and normal stroma and constructed a PPI network of these DEGs. From the PPI network, we identified eight hub genes which were significantly upregulated in CTS and were targeted by the downregulated DEmiRs. Among the eight hub genes, three (*COL5A2*, *EDNRA*, and *OLR1*) showed significant inverse expression correlations with OS and/or DFS in three CRC cohorts (Figure 3). Moreover, the elevated expression of the three prognostic hub genes (PHGs) was significantly correlated with immunosuppressive signatures in colon cancer (Figure 4). Network analysis showed that many immune-inhibitory signature genes interacted with the PHGs (Figure 5A). For example, *OLR1* interected with *CCL2* and *CD68*. These results suggest that the negative prognostic impact of PHGs in colon cancer may result from the suppressive antitumor immune microenvironment. Collectively, the expression pattern, prognostic effect, immune-inhibitory activity demonstrated the oncogenic effect of *COL5A2*, *EDNRA*, and *OLR1* in colon cancer.

Based on the three hub genes and the DEmiRs regulating them, we established an miRNA–mRNA regulatory network that may contribute to the onset and progression of colon cancer. This network included the interactions of hsa-mir-21-5p–COL5A2, hsa-mir-409-3p–COL5A2, hsa-mir-135b-5p/hsa-mir-495-3p–COL5A2, hsa-mir-21-5p–OLR1, hsa-mir-135b-5p/hsa-mir-495-3p–EDNRA, hsa-mir-21-3p–EDNRA and hsa-mir-224-5p–EDNRA (Figure 5B). It suggests that the hsa-mir-21-5p to COL5A2/OLR1, hsa-mir-135b-5p/hsa-mir-495-3p/hsa-mir-409-3p to COL5A2 and hsa-mir-224-5p/hsa-mir-21-3p/hsa-mir-135b-5p/hsa-mir-495-3p to EDNRA pathways may play substantial roles in regulation of the tumor immune microenvironment in colon cancer.

Our results showed that hsa-miR-21-5p, hsa-mir-21-3p, and hsa-mir-409-3p were downregulated in CTS. However, previous studies revealed that the expression of these miRNAs were upregulated in CRC tissues [44,45,46]. It suggests that these miRNAs may play distinct roles in CRC and its TME.

A major limitation of this study is that the CTS-associated miRNA–mRNA regulatory network identified by in silico analysis has not been proved by experimental validation. Thus, although our findings could provide potentially useful biomarkers for colon cancer diagnosis and prognosis, as well as therapeutic targets, further experimental and clinical validation is needed to translate these findings into clinical application.

## 5. Conclusions

The identification of transcriptional markers and miRNA–miRNA regulatory networks in CTS may provide new biomarkers for colon cancer diagnosis, prognosis, and treatment, as well as insights into the mechanism of tumor immune microenvironment regulation in colon cancer.

## Figures and Tables

**Figure 1 cells-08-01054-f001:**
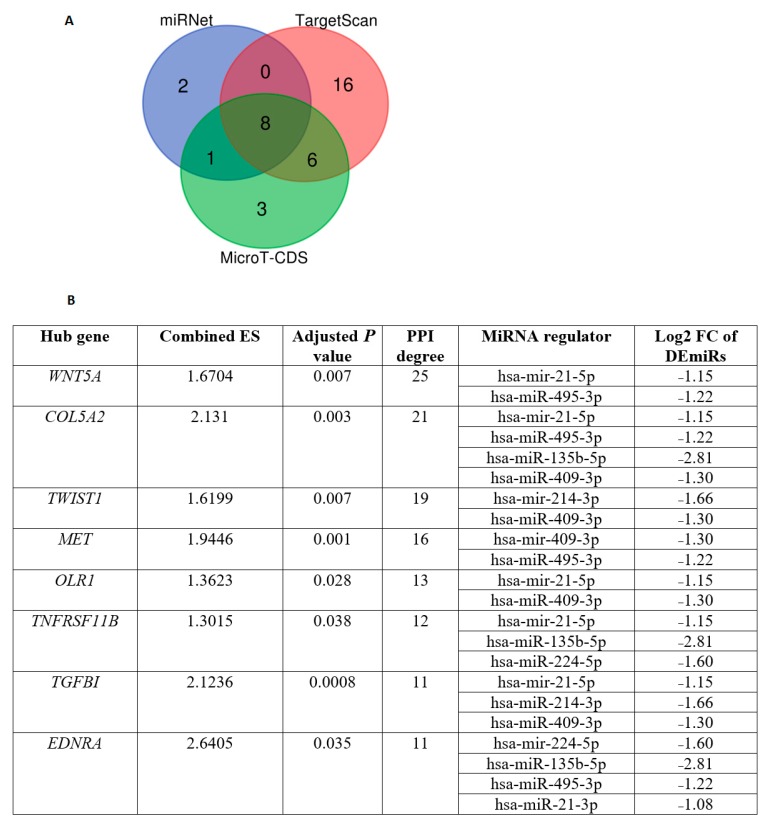
Identification of hub genes targeted by DEmiRs. (**A**) A total of eight hub genes were identified as the targets of downregulated DEmiRs in CTS which were commonly predicted by all three different tools (miRNet [18], MicroT-CDS [38], and TargetScan [22]). (**B**) A detailed description of the eight hub genes and their associated regulatory DEmiRs. ES: effect size. FDR: false discovery rate.

**Figure 2 cells-08-01054-f002:**
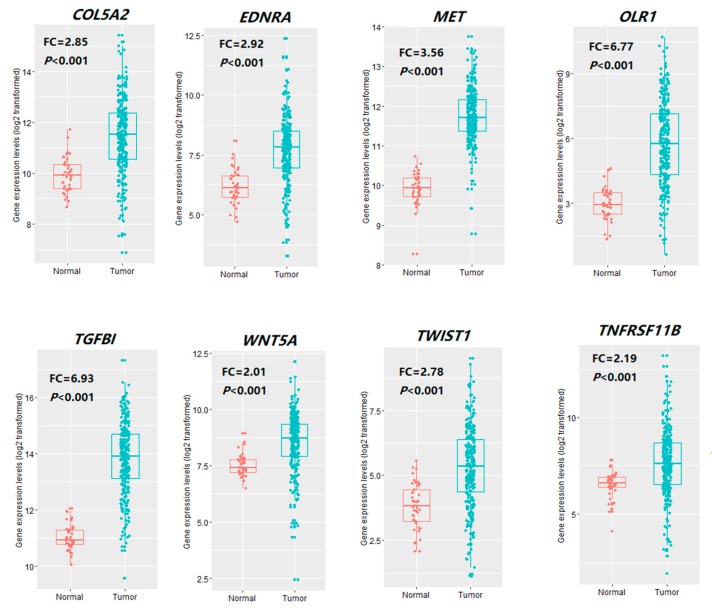
Comparisons of the expression levels of the eight hub genes between TCGA colon cancer and normal tissue. *COL5A2, EDNRA, TGFBI, MET, OLR1, TNFRSF11B, TWIST1,* and *WNT5A* are upregulated in colon cancer versus normal samples (Student′s *t* test, *p* < 0.001). FC: Fold change.

**Figure 3 cells-08-01054-f003:**
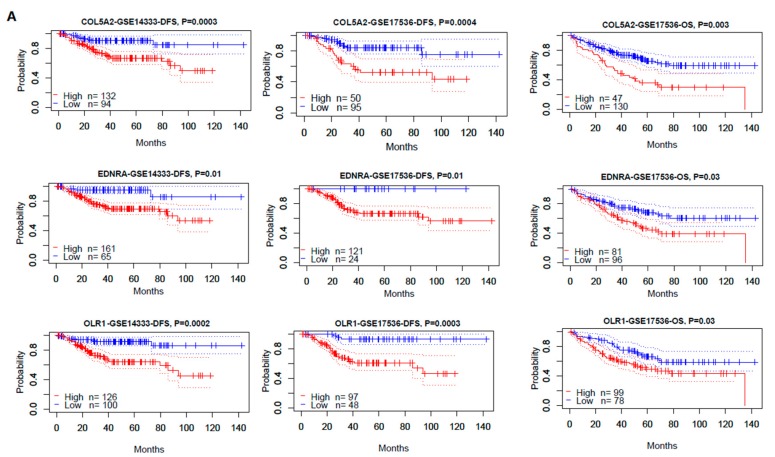
Associations of the expression of hub genes with survival prognosis in colorectal cancer (CRC) patients. (**A**). Kaplan-Meier curves show that three hub genes (*COL5A2*, *EDNRA*, and *OLR1*) had a negative expression correlation with survival prognosis (overall survival (OS) and/or disease-free survival (DFS)) in CRC by Prognoscan (log-rank test, *p* < 0.05). (**B**). Kaplan-Meier curves shows that the elevated expression of four hub genes (*COL5A2*, *EDNRA*, *OLR1*, and *TGFBI*) is associated with a worse DFS prognosis in the TCGA colon cancer cohort by GEPIA [23] (log-rank test, *p* < 0.05).

**Figure 4 cells-08-01054-f004:**
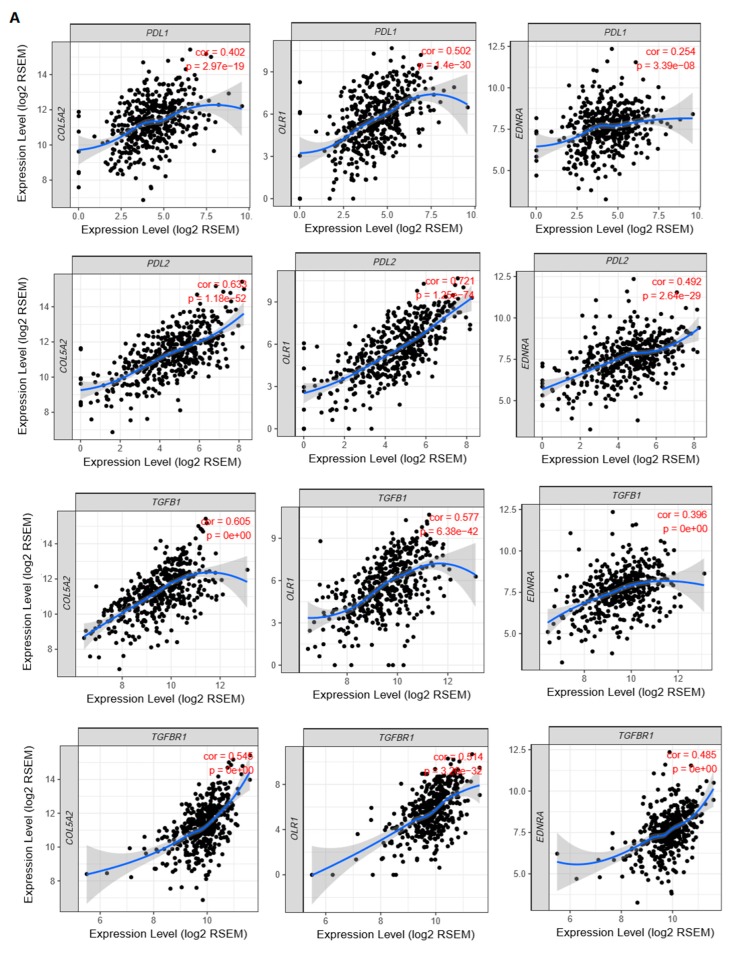
Associations between hub genes and immune signatures in colon cancer. (**A**) Three hub genes (*COL5A2*, *EDNRA*, and *OLR1*) exhibit a significant positive expression correlation with four immune-inhibitory marker genes (*PD-L1*, *PD-L2*, *TGFB1*, and *TGFBR1*). The correlation analyses were performed using TIMER [25]. The Spearman’s correlation test *p*-values (*p*) and correlation coefficients (cor) are shown. RSEM: RNA-Seq by Expectation Maximization [64]. (**B**) The ratios of pro-/anti-inflammatory cytokines are significantly lower in the colon cancers highly expressing the hub genes (expression levels > median) than in those lowly expressing the hub genes (expression levels < median) in the TCGA COAD cohort (Student′s *t* test, *p* < 0.001). The ratio of pro-/anti-inflammatory cytokines in a tumor sample is defined as the ratio of average expression levels (log2-transformed) of their marker genes. The pro-inflammatory cytokines represent the immune-stimulatory signature with marker genes *IFNG*, *IL-1A*, *IL-1B*, and *IL-2*, and the anti-inflammatory cytokines represent the immune-inhibitory signature with marker genes *IL-4*, *IL-10*, *IL-11*, and *TGFB1* [40]. ***, *p* < 0.001.

**Figure 5 cells-08-01054-f005:**
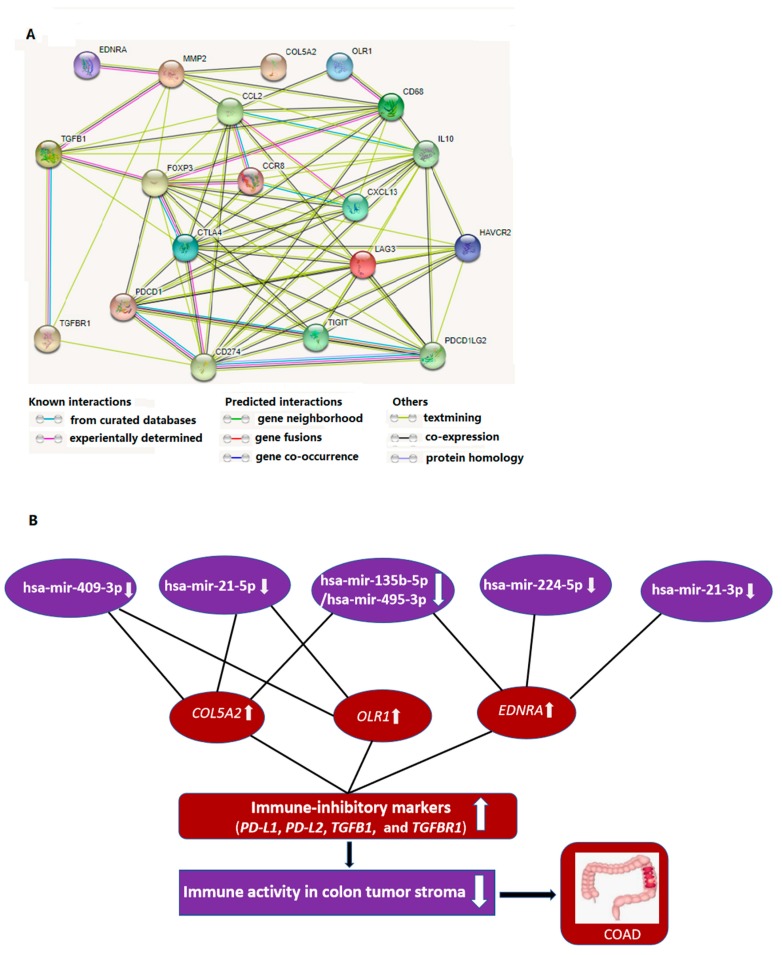
Interaction networks of prognostic hub genes, immune-inhibitory signature genes, and DEmiRs. (**A**) Interaction network of the three prognostic hub genes (*COL5A2*, *EDNRA*, and *OLR1*) and the marker genes of immune-inhibitory signatures (TAMs, Tregs, and T cell exhaustion). This network was constructed by STRING [26]. TAMs: tumor-associated macrophages. Tregs: regulatory T cells. (**B**) The miRNA–mRNA regulatory network of the three prognostic hub genes and their DEmiR regulators that may contribute to the pathogenesis of colon cancer.

**Table 1 cells-08-01054-t001:** Differentially expressed microRNAs (miRNAs) (DEmiRs) between colon tumor stroma (CTS) and normal stroma and the numbers of their target genes commonly identified by three different tools.

miRNA	Log_2_ FC ^a^	FDR ^b^	Number of Target Genes ^c^
hsa-mir-375	2.52	0.03	48
hsa-mir-551b-3p	2.33	0.009	NA ^d^
hsa-mir-513c-5p	2.29	0.005	2
hsa-mir-192-3p	2.10	0.038	15
hsa-mir-215-5p	2.01	0.01	27
hsa-mir-192-5p	1.96	0.01	37
hsa-mir-378a-3p	1.76	0.01	5
hsa-mir-194-5p	1.74	0.04	19
hsa-mir-29c-3p	1.61	0.01	91
hsa-mir-498	1.52	0.009	28
hsa-mir-194-3p	1.46	0.03	8
hsa-mir-29c-5p	1.38	0.008	NA
hsa-mir-10b-5p	1.31	0.01	30
hsa-mir-338-3p	1.31	0.01	18
hsa-mir-139-5p	1.28	0.01	7
hsa-mir-512-3p	1.09	0.01	30
hsa-miR-768-3p	1.09	0.02	NA
hsa-mir-135b-5p	−2.81	0.009	19
hsa-mir-214-3p	−1.66	0.02	38
hsa-mir-224-5p	−1.60	0.01	25
hsa-mir-409-3p	−1.30	0.04	8
hsa-mir-495-3p	−1.22	0.01	43
hsa-mir-21-5p	−1.15	0.04	87
hsa-mir-21-3p	−1.08	0.01	6

^a^ FC: fold change; ^b^ FDR: false discovery rate; ^c^ The number of miRNA target genes which were commonly identified by miRNet [18], MicroT-CDS [38], and TargetScan [22]. ^d^ NA: not available.

**Table 2 cells-08-01054-t002:** The most significant 10 upstream transcription factors (TFs) regulating the upregulated DEmiRs and the most significant 10 upstream TFs regulating the downregulated DEmiRs.

Change of DEmiRs	Upstream TFs	*p*-Value (Hypergeometric Test)
Upregulated	EGR1	6.50 × 10^−37^
	SP1	5.36 × 10^−29^
	SP4	6.01 × 10^−26^
	NKX6-1	5.19 × 10^−22^
	POU2F1	3.44 × 10^−21^
	MEF2A	3.64 × 10^−18^
	RREB1	1.41 × 10^−17^
	ZFP161	1.94 × 10^−16^
	NFIC	2.61 × 10^−15^
	ONECUT1	1.90 × 10^−14^
Downregulated	SP4	1.30 × 10^−17^
	SP1	3.82 × 10^−17^
	EGR1	3.85 × 10^−15^
	HOXA5	1.43 × 10^−8^
	PDX1	1.92 × 10^−8^
	KLF7	3.18 × 10^−8^
	RORA	3.62 × 10^−8^
	POU2F1	3.98 × 10^−8^
	TCF3	9.19 × 10^−8^
	FOXD3	3.27 × 10^−7^

**Table 3 cells-08-01054-t003:** Expression correlations between three hub genes and marker genes of immune-inhibitory signatures evaluated by TIMER [25].

Immune Signature	Marker Gene	*COL5A2*		*EDNRA*		*OLR1*	
		cor	*p*	cor	*p*	cor	*p*
Monocyte	*CD86*	0.629	7.81 × 10^−52^	0.492	2.47 × 10^−29^	0.78	8.21 × 10^−95^
	*CD115*	0.601	<1 × 10^−100^	0.448	<1 × 10^−100^	0.618	1.34 × 10^−49^
TAM	*CCL2*	0.645	<1 × 10^−100^	0.593	<1 × 10^−100^	0.686	6.35 × 10^−65^
	*CD68*	0.42	<1 × 10^−100^	0.243	1.47 × 10^−7^	0.494	1.59 × 10^−29^
	*IL10*	0.427	1 × 10^−21^	0.359	2.15 × 10^−15^	0.537	1.61 × 10^−35^
M2 Macrophage	*CD163*	0.667	2.9 × 10^−60^	0.481	6.39 × 10^−28^	0.741	5.84 × 10^−81^
	*VSIG4*	0.588	<1 × 10^−100^	0.462	<1 × 10^−100^	0.729	3.86 × 10^−77^
	*MS4A4A*	0.575	1.34 × 10^−41^	0.475	4.2 × 10^−27^	0.74	1.92 × 10^−80^
Th1	*T-bet*	0.321	2.02 × 10^−12^	0.175	1.73 × 10^−4^	0.369	3.39 × 10^−16^
	*IFN-γ*	0.176	1.55 × 10^−4^	0.1	3.32 × 10^−2^	0.325	9.92 × 10^−13^
	*TNF-α*	0.314	6.3 × 10^−12^	0.248	7.32 × 10^−8^	0.375	9.92 × 10^−17^
Treg	*FOXP3*	0.496	<1 × 10^−100^	0.358	3.57 × 10^−15^	0.468	2.79 × 10^−26^
	*CCR8*	0.537	1.61 × 10^−35^	0.406	1.34 × 10^−19^	0.535	2.51 × 10^−35^
	*TGFB1*	0.605	<1 × 10^−100^	0.396	<1 × 10^−100^	0.577	6.38 × 10^−42^
T cell exhaustion	*PD-1*	0.268	5.8 × 10^−9^	0.106	2.39 × 10^−2^	0.333	2.6 × 10^−13^
	*CTLA4*	0.375	9.02 × 10^−17^	0.274	2.59 × 10^−9^	0.43	4.53 × 10^−22^
	*LAG3*	0.258	2.44 × 10^−8^	0.109	1.99 × 10^−2^	0.359	2.28 × 10^−15^
	*TIM-3*	0.625	<1 × 10^−100^	0.469	<1 × 10^−100^	0.788	3.58 × 10^−98^
	*TIGIT*	0.408	8.6 × 10^−20^	0.243	1.36 × 10^−7^	0.493	1.8 × 10^−29^
	*CXCL13*	0.365	7.39 × 10^−16^	0.245	1.04 × 10^−7^	0.494	1.54 × 10^−29^
	*LAYN*	0.766	1.75 × 10^−89^	0.636	2.4 × 10^−53^	0.7	1.08 × 10^−68^

cor: Spearman’s rank correlation coefficient; *p*: *p*-value (Spearman’s correlation test).

**Table 4 cells-08-01054-t004:** Eight FDA-approved drugs potentially targeting three hub genes.

Drug	Target Gene	Interaction Types	Score ^d^
ambrisentan	*EDNRA*	antagonist	9
aspirin	*EDNRA*	NA	2
guanfacine	*EDNRA*	antagonist	1
cefadroxil	*EDNRA*	antagonist	1
bosentan	*EDNRA*	antagonist	11
macitentan	*EDNRA*	antagonist	7
collagenase clostridium histolyticum	*COL5A2*	NA	1
ocriplasmin	*COL5A2*	NA	1

^d^ The score is the number of database sources and/or PubMed references supporting the drug-gene interaction.

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
