# Peer review of "Identification of Transcriptional Markers and microRNA–mRNA Regulatory Networks in Colon Cancer by Integrative Analysis of mRNA and microRNA Expression Profiles in Colon Tumor Stroma"

_cells, 2019, doi:10.3390/cells8091054_

Round 1

Reviewer 1 Report

In the present study, using existing gene expression data the authors have identified transcriptional markers and miRNA-mRNA regulatory networks in colon cancer. Here, the authors have performed an extensive computational analysis and the study provides some novel insights into the pathophysiology of colon cancer. A major limitation of the study is the lack of experimental validation at any level, which significantly reduces the impact of their finding.

Figure 1A and B: What does ‘percentage of gene’ refers to? Percentage of all genes or all miR genes or just DEmiRs. Please clarify.

Figure 1C shows predicted target genes of the DEmiRs. Although the figure looks nice, it is not very informative. Could you also please include this information as a supplementary table.

Figure 2A: It is hard to read the information in this figure. Could you please increase the font size or provide this information in a supplementary table as well?

What was the rationale behind using a melanoma dataset in a colon cancer study?

Based on your data it looks like hsa-miR-21-5p plays a crucial role in regulating the hub genes, and it is down-regulated in your data. However, based on published data it is upregulated in colorectal cancer and several other types of cancers. Also, miR-21-5p has been shown to promote proliferation, invasion and metastasis. How would you explain these differences between your study and other published studies? Please discuss this in the ‘discussion’.

Stroma consists of various cell types including fibroblasts, endothelial cells and immune cells.   I would recommend using existing single-cell gene expression data to determine the cell types that express your hub genes.

Author Response

A major limitation of the study is the lack of experimental validation at any level, which significantly reduces the impact of their finding.

Response: We thank the reviewer for this comment. In this study, we focused on the bioinformatics analysis to add novel findings on this topic. Certainly, we agree that experimental validation may add value in this study. This will be one important direction for our future work.

Figure 1A and B: What does ‘percentage of gene’ refers to? Percentage of all genes or all miR genes or just DEmiRs. Please clarify.

Response:  We thank the reviewer for this comment. We have replaced this figure with a table (Table 2) to more clearly present the data. Accordingly, we have removed the ‘percentage of gene’ data because we found that this data is not necessary.

Figure 1C shows predicted target genes of the DEmiRs. Although the figure looks nice, it is not very informative. Could you also please include this information as a supplementary table.

Response:  We have removed Figure 1C and included its information as a supplementary table (Supplementary Table S1).

It is hard to read the information in this figure. Could you please increase the font size or provide this information in a supplementary table as well?

Response:  We have removed Figure 2A and included its information in a supplementary table (Supplementary Table S8).

What was the rationale behind using a melanoma dataset in a colon cancer study?

Response: We thank the reviewer for this comment. We realized the irrationality of using a melanoma dataset in a colon cancer study and thus removed the related data.

Based on your data it looks like hsa-miR-21-5p plays a crucial role in regulating the hub genes, and it is down-regulated in your data. However, based on published data it is upregulated in colorectal cancer and several other types of cancers. Also, miR-21-5p has been shown to promote proliferation, invasion and metastasis. How would you explain these differences between your study and other published studies? Please discuss this in the ‘discussion’.

Response:  We thank the reviewer for this suggestion. Following the suggestion, we have added a discussion about this issue in the Discussion section.

Stroma consists of various cell types including fibroblasts, endothelial cells and immune cells. I would recommend using existing single-cell gene expression data to determine the cell types that express your hub genes.

Response:  We thank the reviewer for this suggestion. Following the suggestion, we found a stromal fibroblast cell gene expression data and performed the related analysis (see Supplementary Table S9).

Reviewer 2 Report

Uddin and coworkers present the manuscript entitled “Identification of Transcriptional Markers and miRNA-mRNA Regulatory Networks in Colon Cancer by Integrative Analysis of mRNA and miRNA Expression Profiles in Colon Tumor Stroma”. This is a interesting  bioinformatic based-study of published datasets. It was well conducted, well written, and conclusions are based in their results without overinterpretation. However, at the same time study showed some flaws. Thus, before potential publication in Cells several concerns should be fully addressed.

Abstract

Authors should define what a “transcriptional marker“ is ?. 

Please rewrite or better explain the sentence or better explain “Furthermore, we identified the transcriptional markers which were hub genes “Furthermore, we identified the transcriptional markers which were hub genes in the protein-protein interaction (PPI) network of DEGs and were targeted by DEmiRs”, as it’s very confusing.

The Background section is very poor. Please complete the section and mention the related published studies on miRNAs/mRNAs in colorectal and other types of human cancers.

Results

Authors used previously published datasets. That’s mean that the data reported here is not original. This arise a lot problems. First, authors should validate the published data by RT-PCR for a limited number of mirnas and mRNAs (ideally 10 mirnas and 10 mRNAs), before continue with bioinformatic analysis of published data by other research groups.

Why authors did not generated its own data in a Chinese population ? Please explain and convince about this relevant question. What’s is the rationale for the use of published data from other geographic locations different to China ? Data obtained in this study will be relevant for Chinese population?

Authors claim “We identified 7 downregulated and 17 upregulated DEmiRs in CTS relative to colon normal stroma (Table 1)”, and show in Table 1 the predicted gene targets using mirNET software. To obtain more robust data, predictions of microRNAs gene targets should be performed by at least 3 different software’s, including TargetScan. Please complete the analysis.

Figure 1. Upstream transcription factors (TFs) regulating DEmiRs and DEmiR-gene regulatory network. A. In figures 1A-B please corrects “Transcrition factors”. Also, figure is very confusing, what’s mean the yellow and red bars?. This is graphic is showing the number of genes that are transcriptional factors?. I understand in figure 1A that 7.2% of total modulate genes were ONeCUT1... it’s that true ? The same for the other genes and data in bars. Please better explains, or remove the figure as it’s very confusing.

Figure 1C. The DEmiR-gene regulatory network. This figure is unreadable, I can’t see the targets genes of mirnas, thus is not informative and should be removed or displayed in a different way.

Figure 2. Identification of hub genes targeted by DEmiRs. Please include the fold change for miRNAs and mRNAs in 2 new columns in the table 2C.

Figure 6. Association of the expression of hub genes with immunotherapy response in a cancer (melanoma) cohort (Hugo cohort [24]) receiving anti-PD-1/PD-L1 immunotherapy. No proper correlations could be drawn between data obtained in melanoma and the datasets analyzed here in colon cancer, as they are very different types of cancer, thus I suggest removing this data from manuscript.

Finally, no English grammar corrections are needed.

Author Response

Abstract a) Authors should define what a “transcriptional marker“ is ?. 

Response: We thank the reviewer for this suggestion. Following this suggestion, we have given a definition of "transcriptional marker" in Abstract.

b) Please rewrite or better explain the sentence or better explain “Furthermore, we identified the transcriptional markers which were hub genes “Furthermore, we identified the transcriptional markers which were hub genes in the protein-protein interaction (PPI) network of DEGs and were targeted by DEmiRs”, as it’s very confusing.

Response: We have rewritten this sentence.

The Background section is very poor. Please complete the section and mention the related published studies on miRNAs/mRNAs in colorectal and other types of human cancers.

Response: We thank the reviewer for this suggestion. We have revised the Background section to make it more complete.

Results a) Authors used previously published datasets. That’s mean that the data reported here is not original. This arise a lot problems. First, authors should validate the published data by RT-PCR for a limited number of mirnas and mRNAs (ideally 10 mirnas and 10 mRNAs), before continue with bioinformatic analysis of published data by other research groups.

Response: We thank the reviewer for the comments and suggestions. In this study, we focused on the bioinformatics analysis of public data to add novel findings on this topic. We agree that experimental validation is important to perform this kind of studies. It should be a priority in our future studies.

b) Why authors did not generated its own data in a Chinese population? Please explain and convince about this relevant question. What’s is the rationale for the use of published data from other geographic locations different to China? Data obtained in this study will be relevant for Chinese population?

Response: This work is a meta-analysis of published datasets. These datasets are not Chinese population relevant since we could not find any Chinese population-associated colon tumor/normal stroma gene expression profiling datasets publically available. However, we agree that a related study of Chinese population would be interesting and meaningful. This is a good direction for our future studies.

c) Authors claim “We identified 7 downregulated and 17 upregulated DEmiRs in CTS relative to colon normal stroma (Table 1)”, and show in Table 1 the predicted gene targets using mirNET software. To obtain more robust data, predictions of microRNAs gene targets should be performed by at least 3 different software’s, including TargetScan. Please complete the analysis.

Response: We thank the reviewer for this suggestion. Following the suggestion, we performed the prediction of gene targets of microRNAs using three different softwares (miRNet, microT-CDS and TargetScan) and presented the results in Supplementary Tables S1-3.

d) Figure 1. Upstream transcription factors (TFs) regulating DEmiRs and DEmiR-gene regulatory network. A. In figures 1A-B please corrects “Transcrition factors”. Also, figure is very confusing, what’s mean the yellow and red bars?. This is graphic is showing the number of genes that are transcriptional factors? I understand in figure 1A that 7.2% of total modulate genes were ONeCUT1... it’s that true? The same for the other genes and data in bars. Please better explains, or remove the figure as it’s very confusing.

Response: Following the suggestion, we have removed Figure 1 and put it information in a new table (Table 2). Accordingly, we have removed the ‘percentage of gene’ data because we found that this data is not necessary.

e) Figure 1C. The DEmiR-gene regulatory network. This figure is unreadable, I can’t see the targets genes of mirnas, thus is not informative and should be removed or displayed in a different way.

Response: Following the suggestion, we have removed Figure 1C and put its information in Supplementary Table S1.

f) Figure 2. Identification of hub genes targeted by DEmiRs. Please include the fold change for miRNAs and mRNAs in 2 new columns in the table 2C.

Response: Following the suggestion, we included the fold change for miRNAs in a new colum. For mRNAs, because this was a meta-analysis, we did not include their fold change data. We think that the equivalent to combined effect size (ES) could be a good alternative to the fold change in this case.

g) Figure 6. Association of the expression of hub genes with immunotherapy response in a cancer (melanoma) cohort (Hugo cohort [24]) receiving anti-PD-1/PD-L1 immunotherapy. No proper correlations could be drawn between data obtained in melanoma and the datasets analyzed here in colon cancer, as they are very different types of cancer, thus I suggest removing this data from manuscript.

Response: Following the suggestion, we have removed this data from manuscript.

Round 2

Reviewer 1 Report

Figure 1 does not show a PPI network, please re-write the figure legend to make it clear.

Author Response

Figure 1 does not show a PPI network, please re-write the figure legend to make it clear.

Response: We thank the reviewer for this comment. We have re-written the figure legend to make it clear.

Reviewer 2 Report

Authors have successfully replied all my concerns, thus I strongly suggested to accept the manuscript for publication in its actual form.

Author Response

We thank the reviewer for the positive response.